# Reversible control over the distribution of chemical inhomogeneities in multiferroic BiFeO$_3$

M. Müller[1], B. Yan [1], H. Ko[1], Y.-L. Huang [2,3,4], H. Lu [5,6], A. Gruverman [5,6], R. Ramesh[3,4,7,8,9], M. D. Rossell [10], M. Fiebig [1] & M. Trassin [1] ✉

Despite the appeal of flawless order, semiconductor technology has demonstrated that implanting inhomogeneities into single-crystalline materials is pivotal for modern electronics. However, the influence of the local arrangement of chemical inhomogeneities on the material's functionalities is under-explored. In this work, we control the distribution of chemical inhomogeneities in La$^{3+}$-substituted ferroelectric BiFeO$_3$ thin films. By means of a stress- and composition-driven phase transition, we trigger the formation of a lattice of La$^{3+}$-rich and La$^{3+}$-poor layers. This ordering correlates with the emergence of an antipolar phase. An electric field restores the original ferroelectric phase and re-randomizes the distribution of the La$^{3+}$ inhomogeneities. Leveraging these insights, we tune the polar/antipolar phase coexistence to set the net polarization of La$_{0.15}$Bi$_{0.85}$FeO$_3$ to any desired value between its saturation limits. Finally, we control the net polarization response in device-compliant capacitor heterostructures to show that inhomogeneity-distribution control is a valuable tool in the design of functional oxide electronics.

Introducing chemical inhomogeneities into crystalline materials has been crucial for technological advancement[1,2]. This is particularly evident in silicon technology, where the integration of impurity atoms constitutes the very essence of semiconductor functionalities. However, introducing these inhomogeneities with uncontrolled, typically homogeneous distribution is no longer sufficient to sustain the current technological development[3,4]. The discovery that important functionalities, such as ferroic order[5,6], superconductivity[7,8], magnetoresistance[9], and even CO$_2$ photoreduction[10] often depend on the configuration of inhomogeneities in the crystal structure underlines this point. As a result, the focus of materials refinement has been shifting from random toward ordered insertion of inhomogeneities

during synthesis. In a complementary development, there is an emerging perspective of reversibly tuning material properties without inserting or removing inhomogeneities, but solely by rearranging them. The potential of this method is far from being exploited, however, since inhomogeneities typically remain localized after material fabrication, which inhibits post-synthesis functionality control[11–14].

A class of functional materials prone to chemical inhomogeneities are ferroelectric perovskite oxides[15–18], materials that are discussed as critical components of next-generation information technologies[4,19,20]. Recent studies on BaTiO$_3$, Pb[Zr$_{0.2}$Ti$_{0.8}$]O$_3$, and SrTiO$_3$ have demonstrated active control over the arrangement of anionic

[1]Department of Materials, ETH Zurich, Zurich, Switzerland. [2]Department of Materials Science and Engineering, National Yang Ming Chiao Tung University, Hsinchu, Taiwan. [3]Department of Materials Science and Engineering, University of California Berkeley, Berkeley, USA. [4]Materials Science Division, Lawrence Berkeley Laboratory, Berkeley, Berkeley, USA. [5]Department of Physics and Astronomy, University of Nebraska, Lincoln, NE, USA. [6]Nebraska Center for Materials and Nanoscience, University of Nebraska, Lincoln, NE, USA. [7]Department of Physics, University of California Berkeley, Berkeley, USA. [8]Department of Materials Science and Nanoengineering, Rice University, Houston, USA. [9]Department of Physics and Astronomy, Rice University, Houston, USA. [10]Electron Microscopy Center, Empa, Swiss Federal Laboratories for Materials Science and Technology, Dübendorf, Switzerland. ✉e-mail: morgan.trassin@mat.ethz.ch

inhomogeneities such as oxygen off-stoichiometry[21–23]. The associated tunability of electrical conduction and polarization kinetics opens up a new degree of freedom in materials engineering. Since cations determine the ferroelectric behavior, establishing control over the distribution of cationic inhomogeneities, a largely unexplored frontier, is of great importance[24].

Here, we reversibly create and annihilate ordered arrangements of cationic inhomogeneities in ferroelectric single-layer $La_{0.15}Bi_{0.85}FeO_3$ thin films. Specifically, using a diamond-coated scanning-probe tip, we apply a compressive force to create layer-dependent variations in the $La^{3+}$ concentration. Strikingly, the local pressure-induced transition from polar to antipolar order correlates with the arrangement of the $La^{3+}$ inhomogeneities in the lattice. By tuning the magnitude of the mechanical force, we manipulate the polar/antipolar phase coexistence to create quasi-continuous net polarization between zero and the saturation polarization of $La_{0.15}Bi_{0.85}FeO_3$. Finally, we demonstrate the device potential of the reversible polar-to-antipolar transition and of $La^{3+}$ distribution control in capacitor heterostructures. This outlines how the control of inhomogeneity distribution can be utilized to adapt the properties of functional oxides post-growth in support of the development of novel multifunctional electronics components.

For this study, we chose $La^{3+}$-substituted multiferroic $BiFeO_3$ as the model system. $La^{3+}$ inhomogeneities substitute for the $Bi^{3+}$ ions, which is responsible for the ferroelectric order[24–26]. Their substitution has enabled magnetoelectric operation at voltages close to technological requirements[27]. The proximity of multiple structural phases in $BiFeO_3$ with respect to the $La^{3+}$ concentration[14,28–30] or external stimuli[31–36] renders $La^{3+}$-substituted films the ideal platform to study the relation between chemical inhomogeneities, their distribution, and the rich functionalities of oxide thin films. The existence of a morphotropic polar-to-antipolar transition for a $La^{3+}$ substitution level of 15% in the bulk form[37] motivates the study of $La_{0.15}Bi_{0.85}FeO_3$ thin films.

## Results and discussion

We grew epitaxially strained $(001)_{p.c.}$-oriented $La_{0.15}Bi_{0.85}FeO_3$ films with a thickness of 100 nm on a 14-nm-thick $SrRuO_3$ buffer layer on top of $(110)_o$-oriented single-crystalline $DyScO_3$ substrates using pulsed laser deposition. Here, we chose the lattice matching $DyScO_3$ substrate motivated by the studies reporting high crystalline quality and excellent properties of the $BiFeO_3$ thin film system[20,26,38]. The strain state, orientation, and thickness of the films were characterized using X-ray diffraction, see supplementary information, Fig. S1. The subscripts "p.c." and "o" refer to the pseudocubic and orthorhombic lattices of $BiFeO_3$ and $DyScO_3$, respectively.

We start the investigation by probing the distribution of the $La^{3+}$ ions in the pristine $La_{0.15}Bi_{0.85}FeO_3$ films with high-angle annular dark-field (HAADF) scanning transmission electron microscopy (STEM). The intensity in the HAADF-STEM micrographs scales approximately with the square of the atomic number, $Z$[39,40], thereby allowing us to differentiate between $Bi^{3+}$ and $La^{3+}$ ions located at the $A$-site of the $ABO_3$ perovskite structure. The HAADF-STEM image of our pristine film and the superimposed normalized intensities associated with the $A$-site ($Bi^{3+}$ and $La^{3+}$, noted as $Bi^{3+}/La^{3+}$) atomic columns are plotted in Fig. 1a. The regular intensity pattern and the uniformity of the averaged vertical and horizontal line profiles confirm the expected homogeneous distribution of the $La^{3+}$ ions in the pristine $La_{0.15}Bi_{0.85}FeO_3$ films.

The impact of the homogeneous $La^{3+}$ distribution on the functionality of the films is investigated using atomic force microscopy (AFM) and piezoresponse force microscopy (PFM). The topography of the film, as well as out-of-plane and in-plane polarization-domain configurations, are shown in Fig. 1b–d. The films exhibit a smooth topography with unit-cell-high step terraces. Moreover, the comparison of the contrast in vertical PFM (VPFM) between the pristine, upward- and downward-polarized regions after PFM tip-induced poling indicates a mostly downward-oriented polarization in the out-of-plane direction in the pristine state, and distinct three piezoresponse levels in lateral PFM (LPFM) images, representative of all

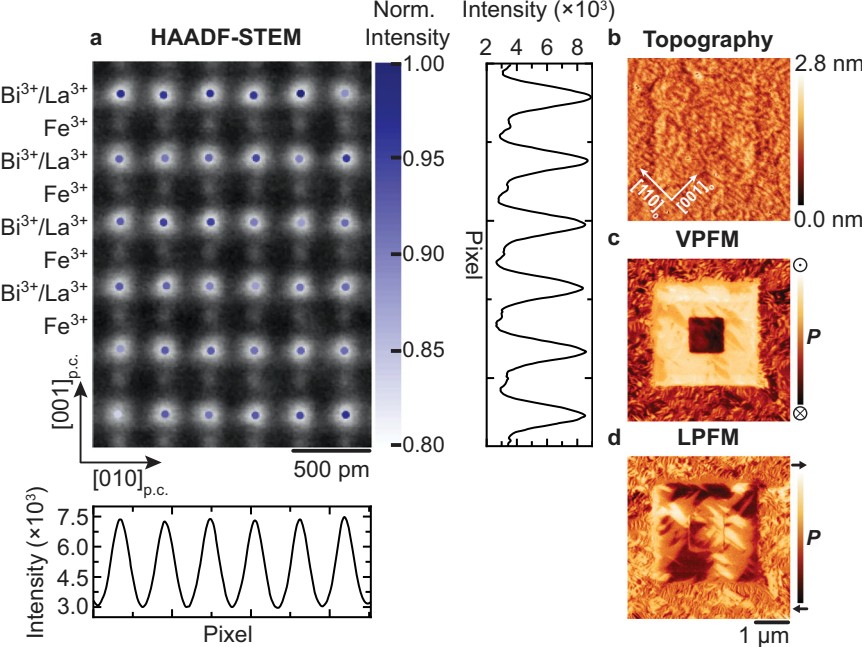

**Fig. 1 | Distribution of $La^{3+}$ inhomogeneities and associated properties of epitaxial $La_{0.15}Bi_{0.85}FeO_3$ thin films. a** HAADF-STEM micrograph of a pristine $La_{0.15}Bi_{0.85}FeO_3$ film captured along $[100]_{p.c.}$. Superimposed on the HAADF-STEM image are the STEM intensities normalized to the maximum intensity of the A-site (labeled $Bi^{3+}/La^{3+}$) atomic columns plotted at their fitted coordinates. The averaged line profiles in vertical and horizontal directions demonstrate the homogeneous distribution of the $La^{3+}$ ions. **b–d** Topography, VPFM, and LPFM micrographs of a pristine $La_{0.15}Bi_{0.85}FeO_3$ film. A local out-of-plane polarization reversal is induced by the application of −/+ 10 V scanning-probe tip bias. The scanning direction is horizontal in (**b–d**).

four in-plane polarization-domain states of BiFeO$_3$[41]. Hence, the PFM investigation confirms the ferroelectric order in the pristine La$_{0.15}$Bi$_{0.85}$FeO$_3$ films. The La$^{3+}$ substitution causes the in-plane polarization domains to arrange randomly, in comparison to the neatly ordered stripe-domain arrangement in BiFeO$_3$ on (110)$_o$-oriented DyScO$_3$ substrates[26]. Because of this, the net-in-plane polarization of the pristine La$_{0.15}$Bi$_{0.85}$FeO$_3$ is substantially attenuated with respect to pure BiFeO$_3$[26].

In the next step, we tune the distribution of the cationic La$^{3+}$ inhomogeneities. A way to act on the La$^{3+}$ arrangement is indicated by the rich phase diagram BiFeO$_3$ exhibits under hydrostatic pressure[42]. Pure BiFeO$_3$ loses its spontaneous polarization under the influence of mechanical force through a ferroelectric-to-antipolar phase transition. Motivated by the recent reports highlighting correlations between lattice chemistry and polarization in epitaxial systems[43–45], we thus apply a compressive force and study its influence on the La$^{3+}$ distribution in the films.

Using diamond-coated AFM tips with a relatively large apex curvature radius (~100 nm), we applied a local mechanical force up to 140 µN to the film surface while raster-scanning a square-shaped region. Strikingly, after exposure to the compressive force, the HAADF-STEM micrograph shows a reduced intensity at the A-site of the $ABO_3$ perovskite in every other Bi$^{3+}$- and La$^{3+}$-containing plane, see Fig. 2a. This STEM intensity modulation is evident in the line profile extracted along the vertical of the image in Fig. 2a. The results are depicted in Fig. 2b and show two distinct STEM intensity levels of the (001)$_{p.c.}$-planes that alternate along the [001]$_{p.c.}$-direction. They indicate a periodic modulation of the average atomic number when passing from one layer to another. We thus conclude that the applied compressive force triggered an alternating enrichment and depletion of the La$^{3+}$ distribution in the films. This is further supported by complementary electron energy-loss spectroscopy and energy dispersive X-ray analysis, see supplementary information Fig. S2a and Table S1.

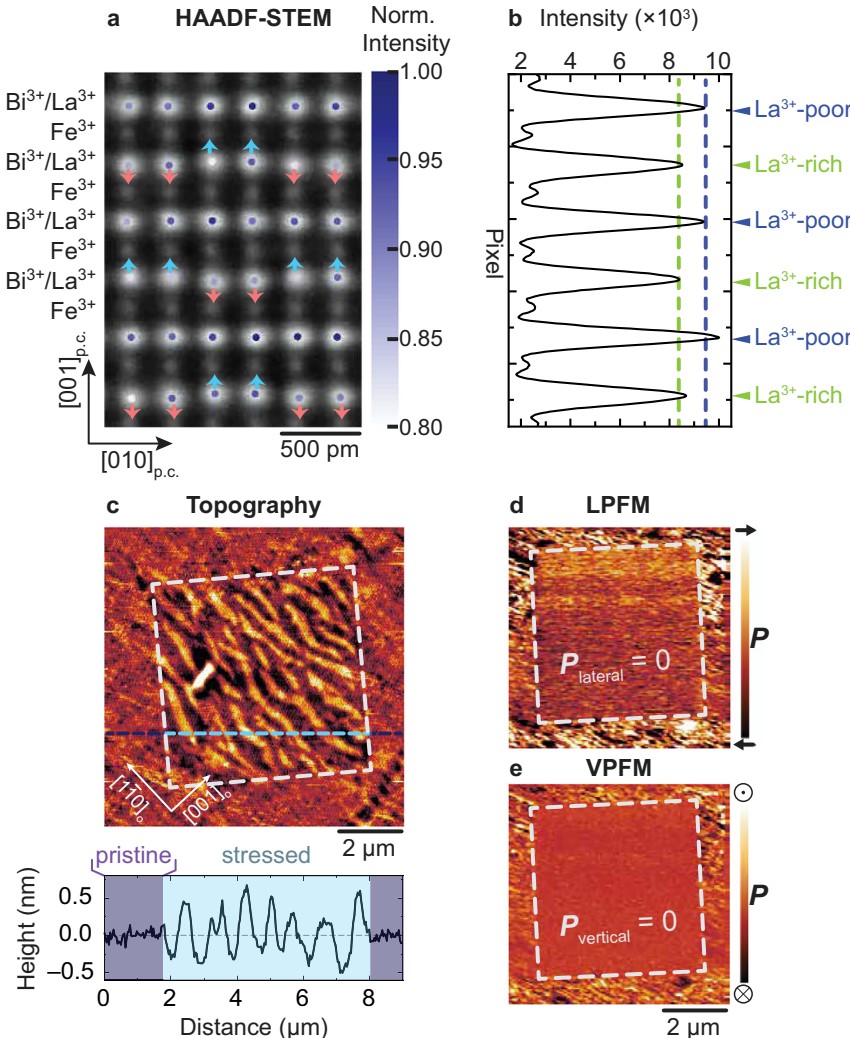

**Fig. 2 | Compressive force-induced rearrangement of La$^{3+}$ inhomogeneities.**
**a** HAADF-STEM micrograph along [100]$_{p.c.}$ of a region that has been exposed to 140 µN of compressive force. Superimposed on the HAADF-STEM image are the STEM intensities normalized to the maximum intensity of the A-site (labeled Bi$^{3+}$/La$^{3+}$) atomic columns plotted at their fitted coordinates. The blue and red arrows indicate the up- and down-shifts of the Bi$^{3+}$/La$^{3+}$ ions, respectively. **b** Averaged vertical line profile of (**a**). The green and dark blue dashed lines indicate the two distinct intensity levels of alternating La$^{3+}$-rich and La$^{3+}$-poor planes, respectively. **c** Topography micrograph demonstrating the surface reorganization of a region after exposure to a compressive force of 140 µN—indicated by a white dashed square—with respect to the pristine region. The line profile below corresponds to the dashed blue line in the micrograph. The purple- and light blue-shaded regions denote the pristine and stressed areas, respectively. **d**, **e** LPFM, and VPFM micrographs of the same region as depicted in (**c**). The antipolar region is marked with a dashed white square.

In addition to the oscillations of the average atomic weight, the $La^{3+}$-enriched layers now reveal an alternating up-up–down-down shift of the $Bi^{3+}/La^{3+}$ ions. The cation shifts are indicated by the blue and red arrows in Fig. 2a. Note that the influence of the cation shifts on the intensity profile in Fig. 2b is subtracted in the analysis shown in the supplementary information Fig. S2b.

The observed structural deformation, i.e., alternating up-up–down-down shift in every other *A*-site plane in Fig. 2a, is characteristic of an antipolar phase with the space group *Pnma*[46,47], see supplementary information, Fig. S3. Additional HAADF-STEM micrographs taken at the boundary between the ferroelectric and antipolar regions demonstrate a clear spatial separation between layered $La^{3+}$-arrangements with antipolar electric-dipole orderings and the pristine phase with randomized $La^{3+}$ distribution, see supplementary information, Fig. S4. Hence, the experimental results show a striking correlation between the lattice chemistry and the polar nature of the $La_{0.15}Bi_{0.85}FeO_3$ unit cell. In ferroelectric $BiFeO_3$, the $Bi^{3+}$ electronic lone pair drives local polar distortions in the ferroelectric unit cell in $BiFeO_3$[24–26]. Hence, the $Bi^{3+}$ substitution with $La^{3+}$ influences the net polar state in the La-$BiFeO_3$ films. At the pressure-induced polar-to-antipolar phase transition in highly $La^{3+}$ substituted $BiFeO_3$ thin films, the reorganization of the atomic displacement within the unit cell may, hence, correlate with a change in lattice chemistry. Here, the identical radii and valence states of $Bi^{3+}$ and $La^{3+}$ cations at play most likely facilitate the reversible cationic redistribution across the phase transition, as highlighted in the report of the different polar regions in the phase diagram of $BiFeO_3/LaFeO_3$ superlattices[48].

In the next step, we investigate the topography and ferroelectric domain configuration of the antipolar region. Using AFM, we observe a transition from an initially flat surface to a corrugated pattern, see Fig. 2c. We attribute this corrugation to the vertical compression and resulting lateral expansion of the $La_{0.15}Bi_{0.85}FeO_3$ lattice in the antipolar *Pnma* phase, see supplementary information, Fig. S4c, d. In addition, the lack of piezoresponse in both the LPFM and VPFM micrographs validates the transition to the antipolar phase in this region, see Fig. 2d and e, respectively.

We tested whether mechanical-damage-induced artifacts can explain the observed AFM/PFM results by studying the dependence of the topographic corrugation on the raster-scanning direction during force application. The results, depicted in supplementary information Figs. S5 and S6 show that neither surface reconstruction[49] nor chemistry changes[48] occur in the regions of interest and that the corrugation always aligns with respect to the $[1\bar{1}0]_o$-axis of the $DyScO_3$ substrate, independent of the diamond-tip scanning direction. The disconnection between scanning direction and topographic features, combined with the HAADF-STEM results, allows us to exclude mechanical-damage-induced artifacts as the cause of the topographic alteration in Fig. 2c and the loss of piezoresponse in Fig. 2d and e.

With the knowledge that a compressive force can rearrange $La^{3+}$ ions in the $La_{0.15}Bi_{0.85}FeO_3$ films, we now investigate the force-magnitude dependence of this process. We applied forces from 21 µN to 139 µN to six different regions of the sample and assessed the degree of the $La^{3+}$ ordering by studying the ferroelectric polarization using LPFM. We observe a progressive loss in polarization-domain contrast, indicative of the steady transition from the ferroelectric to the antipolar phase with increasing force. A selection of the respective LPFM micrographs is depicted in Fig. 3a–d.

We quantify the suppression of the polarization by extracting the width of the LPFM histogram of the stressed-square region and normalizing it to the according value for the pristine region. Note that the LPFM histogram depicts the presence and number of piezoresponse levels associated with in-plane polarization domains, see supplementary information, Fig. S7. The gradual fading of piezoresponse contrast with increasing force can thereby be translated to a quantitative estimate of the extent of the polar-to-antipolar phase transition. The result is depicted in Fig. 3e and corroborates that we have identified a new handle to adjust the macroscopic polarization level of the $La_{0.15}Bi_{0.85}FeO_3$ films quasi-continuously between zero and the saturation value by making use of the pressure-controlled polar-to-antipolar phase transitions. This tunability is an essential feature for multi-level information technology[50–53]. Note that, since the antipolar order is accompanied by an alternating $La^{3+}$ distribution, we conclude that the degree of $La^{3+}$ ordering scales with the applied force, too.

Next, we investigate whether the force-controlled inhomogeneity distribution and the associated variation in spontaneous polarization can be reversed. Therefore, we applied a local electric field in the form of an 8 V bias to a conducting PFM tip and poled a small square within the antipolar region. Strikingly, a cross-sectional HAADF-STEM micrograph of a region treated in this way reveals the return to the homogeneous $La^{3+}$ distribution and back-conversion to the ferroelectric *R3c* phase, see Fig. 4a. This remarkable structural and dielectric reversibility is corroborated by the scanning-probe investigations in

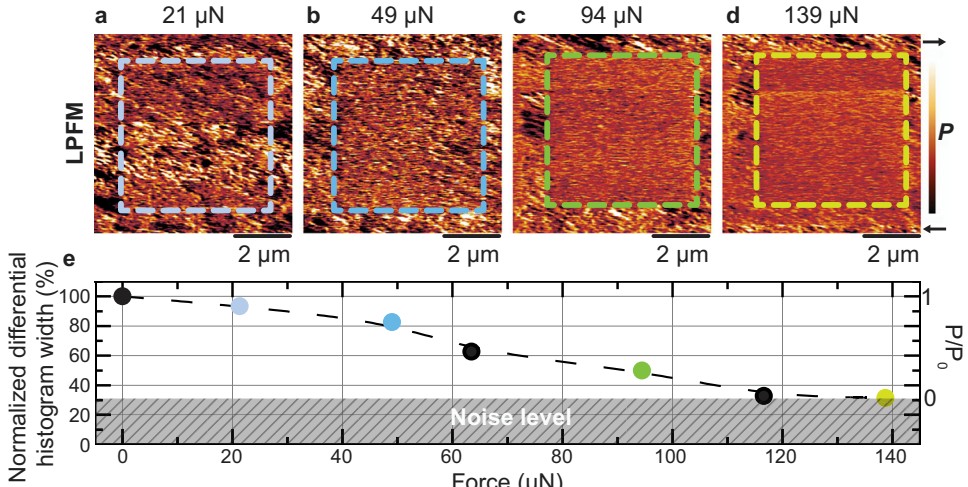

**Fig. 3 | Continuous tuning of phase coexistence. a–d** LPFM micrographs of a region where a compressive force of **a** 21 µN, **b** 49 µN, **c** 94 µN, and **d** 139 µN was applied. The exposed regions are outlined with colored dashed boxes. **e** Piezoresponse histogram width from the treated regions, relative to the histogram width of the surrounding pristine region and associated normalized macroscopic polarization as a function of the applied force. The color of the data points corresponds to that of the dashed boxes in (**a–d**). The LPFM micrographs corresponding to the black data points are not shown.

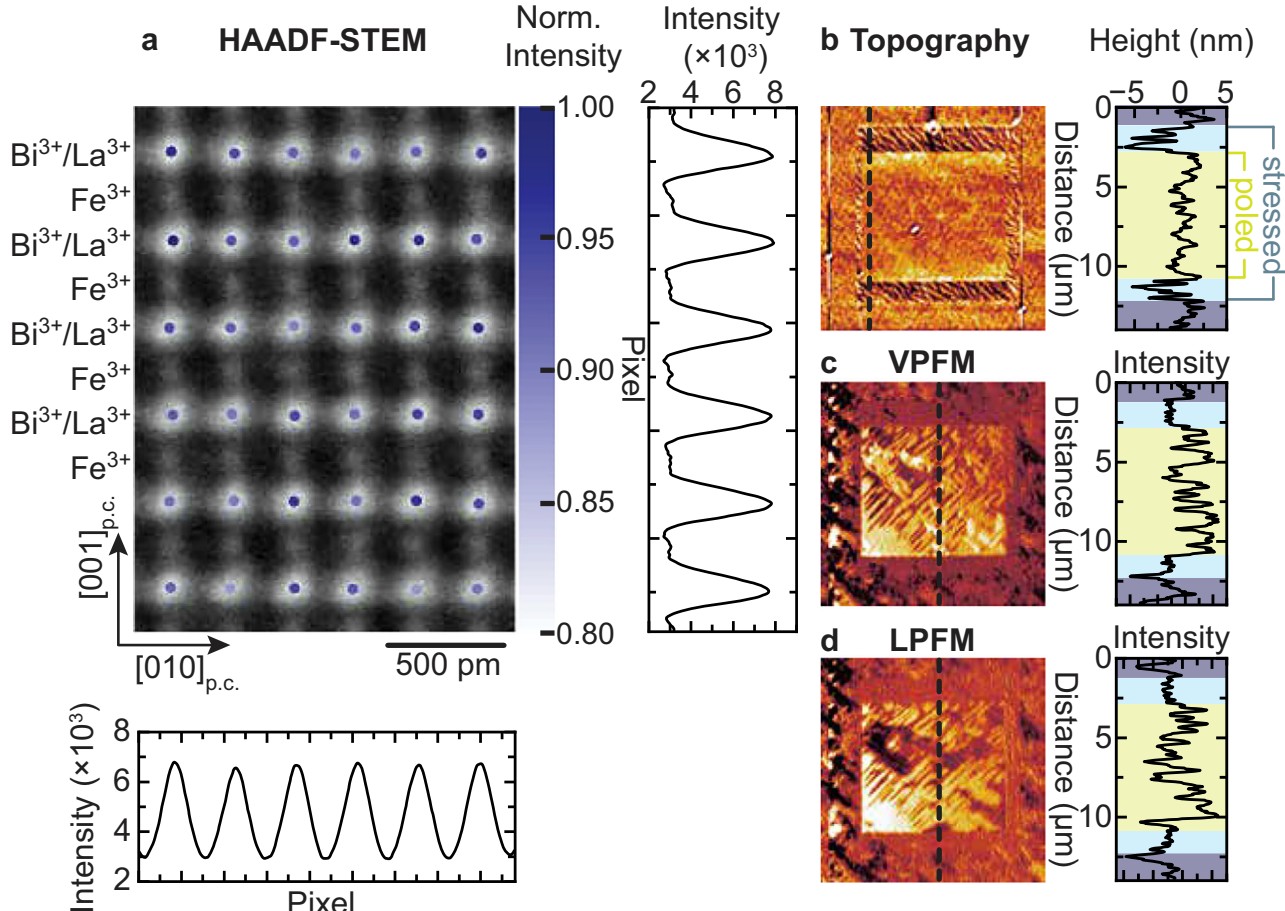

**Fig. 4 | Reversibility of the La³⁺ ordering. a** HAADF-STEM micrograph along [100]_p.c. of a region that had first been exposed to 140 µN compressive force and subsequently to a voltage of 5 V. Superimposed on the HAADF-STEM image are the STEM intensities normalized to the maximum intensity of the A-site (labeled Bi³⁺/La³⁺) atomic columns plotted at their fitted coordinates. The averaged line profiles in vertical and horizontal directions demonstrate the return to the homogeneous distribution of La³⁺ ions. **b–d** Topography, VPFM, and LPFM micrographs of a stressed and subsequently poled region. The associated line profiles correspond to the dashed lines in the micrographs. The shaded regions in the line profiles denote the pristine (purple), stressed (light blue), and subsequently poled (light green) regions.

Fig. 4b–d. We also observe a return to the flat topography of the pristine film in the poled region, see Fig. 4b, and to the original piezoresponse contrast in the PFM micrographs, see Fig. 4c, d. These observations, therefore, demonstrate full reversibility of the force-induced La³⁺ and antipolar order under an electric field. In supplementary information, Fig. S8, we show that this cycle can be repeated multiple times. Finally, we tested the impact of the La³⁺ substitution on the phase conversion, see supplementary Figs. S9 and S10. The polar-to-antipolar phase transition is only achieved for 10 and 15 % La³⁺ substitution levels. Films with 20% La³⁺ substitution already exhibit the antipolar phase in the pristine state (Fig. S10), in agreement with reports on highly La³⁺-substituted BiFeO₃ films⁵⁴. Here, the ferroelectric phase cannot be triggered by an external electric field. Films without La³⁺ substitution, i.e., pure BiFeO₃ films, exhibit ferroelectric order in the pristine state, and the application of local stress does not trigger the emergence of the antipolar phase, as shown in Fig. S9a–c.

For a quantitative analysis of the voltage needed to restore the original ferroelectric state from the antipolar phase, we performed local PFM switching spectroscopy measurements on pristine ferroelectric regions and stressed antipolar regions. Figure 5a shows the topography within the antipolar region before (left panel) and after (right panel) local poling. The white circles, respectively squares indicate the location of the immobile tip during the electric-field cycling, starting with the negative, respectively positive polarity of the tip bias. The clear topography change at these locations demonstrates the

ability to recover the ferroelectric phase at the nanoscale. The PFM switching spectroscopy loops measured in the ferroelectric and antipolar states are shown in Fig. 5b and c, respectively. While the voltage-dependent piezoresponse shows the characteristic hysteretic behavior in the ferroelectric regions, see Fig. 5b, a clear signature of the antipolar phase can be tracked at sub-ferroelectric-coercive field voltage (<2 V) in Fig. 5c. Both the piezoresponse phase and amplitude signals are suppressed in the antipolar phase in the initial poling sequence in Fig. 4c. The transition to the ferroelectric state is not voltage-polarity dependent. Finally, we note that we did not detect any variation of the pressure-induced or poled features over a time period exceeding three months, as documented in supplementary information, Fig. S11.

In order to explore the technological potential of our findings, we finally integrate the La₀.₁₅Bi₀.₈₅FeO₃ films into a device-like architecture. Inspired by the application-relevant integration of BiFeO₃ in ferromagnetic/multiferroic heterostructures [20,27,38,55,56], we deposited circular Co₉₀Fe₁₀ (2 nm) /Pt (2 nm) top electrodes, hence creating a SrRuO₃/La₀.₁₅Bi₀.₈₅FeO₃/Co₉₀Fe₁₀/Pt capacitor.

To probe its ferroelectric polarization non-invasively through the top and bottom electrodes, we used optical second-harmonic generation (SHG). This technique describes the frequency-doubling of electromagnetic waves in materials lacking inversion symmetry⁵⁷. In our experimental geometry, the SHG intensity scales quadratically with the net-in-plane polarization magnitude²⁶,⁵⁸. Hence, we can distinguish the antipolar, centrosymmetric from the ferroelectric, non-

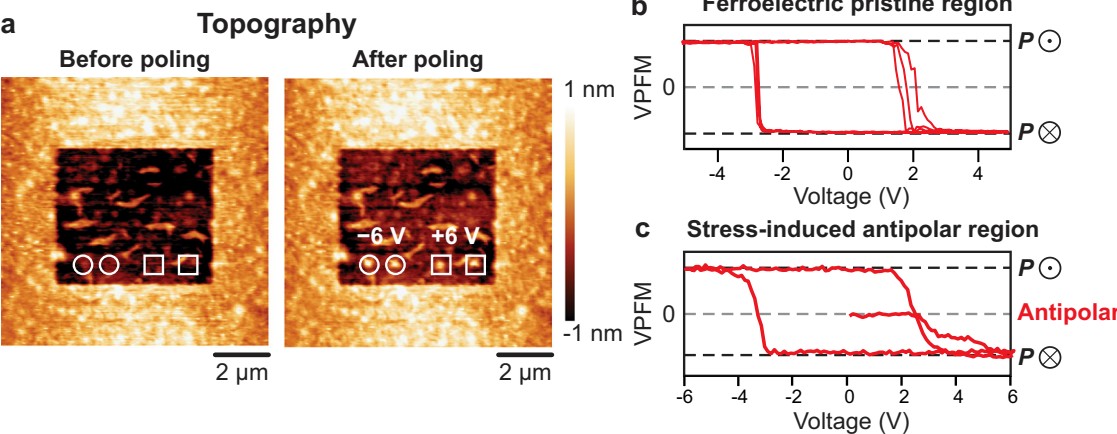

**Fig. 5 | Local control of voltage-driven antipolar-to-ferroelectric phase transition. a** Topography of a stressed region before and after the subsequent electric-field cycling starting with −6 V and +6 V local electric pulses. The poled location starting with negative and positive voltage polarities are indicated by white circles and squares, respectively. **b, c** Representative PFM phase switching spectroscopy loops of the pristine ferroelectric region (**b**) and stressed-induced antipolar region (**c**). Upward- and downward-oriented polarization states are noted as $P\odot$ and $P\otimes$, respectively.

centrosymmetric phase non-destructively by the absence or presence of an SHG signal, respectively.

Figure 6a shows the spatially resolved SHG emission from the capacitor before force application. Note that $10^3$ bipolar voltage pulses were applied to enhance the net-in-plane polarization by reorganizing the randomized pristine domain configuration into a regular pattern of stripe domains[26]. The resulting net in-plane polarization within the capacitor gives rise to a strong SHG signal[26]. (Note that some regions in the outer rim of the capacitor exhibit low SHG intensity. In this area, the in-plane polarization domains remain homogeneously distributed, leading to a weak net in-plane polarization and, thus, negligible SHG emission.) Subsequent application of a compressive force to the top electrode leads to a suppression of the SHG emission, see Fig. 6b. Finally, Fig. 6c shows that application of an electric field to the $La_{0.15}Bi_{0.85}FeO_3$-based capacitor recovers the original SHG intensity.

We have thus evidenced that the phase transition from ferroelectric to antipolar occurs in a device architecture just like in the original $La_{0.15}Bi_{0.85}FeO_3$ films. In turn, we conclude that the $La^{3+}$ ions can be redistributed into a layered arrangement even in capacitor structures. Moreover, we have observed the return of the ferroelectric phase and, thus, our ability to complete a polarization-mediated write-and-erase cycle in a prototypical device architecture.

In striking contrast to previous reports, in which the stabilization of the antipolar *Pnma* phase required elaborate elastic and electrostatic boundary-condition engineering in complex $La_{0.4}Bi_{0.6}FeO_3$/$BiFeO_3$[46] and $TbScO_3$/$BiFeO_3$[47] superlattices, we stabilize the antipolar phase in a single $La_{0.15}Bi_{0.85}FeO_3$ layer directly integrated into a capacitor. This situation suggests that rather than electrostatics, strain, or surface chemistry, the force-dependent distribution of cationic $La^{3+}$ inhomogeneities is originally responsible for the robustness of both the ferroelectric and antipolar phases in the single layers and capacitors. Furthermore, we tested the robustness of the reversible polar-to-antipolar phase conversion on single-crystalline substrates other than $DyScO_3$. The local AFM and PFM images of stressed and subsequently poled regions in films grown on (001)-oriented $SrTiO_3$ demonstrate that the local control on the $La^{3+}$ inhomogeneities and corresponding polar-to-antipolar phase transition is achieved with similar experimental parameters, see supplementary Fig. S12. Hence, the findings reported in this work are not limited to the use of orthorhombic $DyScO_3$ substrates.

In conclusion, we redistribute cationic inhomogeneities in a magnetoelectric oxide material using compressive force and electric fields.

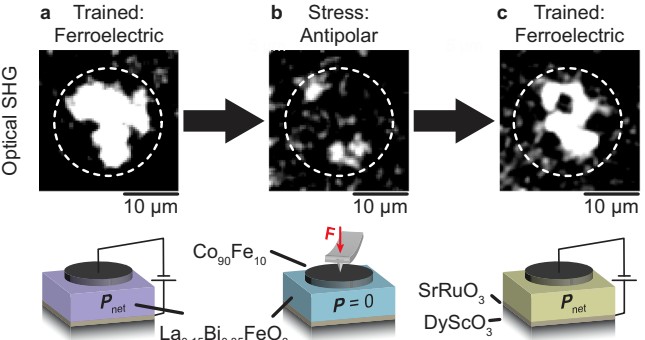

**Fig. 6 | Control over the $La^{3+}$ arrangement and ferroelectric/antipolar phase coexistence in a capacitor device. a–c** Optical SHG images of a $SrRuO_3$/$La_{0.15}Bi_{0.85}FeO_3$/$Co_{90}Fe_{10}$ capacitor after **a** $10^4$ ferroelectric-switching events, **b** subsequent application of 140 μN, and **c** consecutive $10^4$ ferroelectric-switching events. The dashed white circle outlines the electrode region with a diameter of 20 μm. Bright regions correspond to high SHG emission and, therefore, denote the presence of a macroscopic polarization. The regions with low SHG intensity within the capacitor area indicate low net in-plane polarization due to incomplete training. Schematics of the ferroelectric-switching and force-application experiments are depicted below the respective SHG micrograph. The color of the $La_{0.15}Bi_{0.85}FeO_3$ in the sketches denotes its pristine ferroelectric (purple), force-induced antipolar (light blue), and consecutively poled ferroelectric (light green) phases.

In the model system $La_{0.15}Bi_{0.85}FeO_3$, we demonstrate that the rearrangement between randomized and layered distributions of $La^{3+}$ ions enables reversible and continuous interconversion between ferroelectric and antipolar phases. In a step toward multi-level computing technology, we transfer this concept to a device-like capacitor environment. This work on the $BiFeO_3$ material family, the only room-temperature multiferroic material to date and the most promising compound toward the realization of ultra-low-energy consuming logic devices[27], advances the development of future ferroelectric and magnetoelectric memory technology. Here, the continuous tuning of the fraction of antipolar and ferroelectric phases via phase interconversion may enable memristive behavior as well in the magnetoelectric response. While our investigation focused on 100-nm-thick layers, we report the reversible control on the polar state in 20-nm-thick layers in the supplementary information, Fig. S13. Lastly, the scope of controlling

cationic inhomogeneity distributions goes beyond mere polarization engineering, however. We envision multifunctional oxide-based devices whose mechanical, electrical, or optical properties at large can be readily adjusted through cationic inhomogeneity-distribution control.

## Methods

### Sample fabrication

The $La_{0.15}Bi_{0.85}FeO_3/SrRuO_3$ films were grown on single-crystalline $(110)_o$-oriented $DyScO_3$ substrates (CrysTec GmbH) by pulsed laser deposition using a 248 nm KrF excimer laser. The $SrRuO_3$ buffer layer was deposited at 700 °C under 0.15 mbar oxygen partial pressure with a laser fluence of 0.69 J cm$^{-2}$ and a laser repetition rate of 8 Hz (9600 pulses for the growth of 14-nm-thick film). The $La_{0.15}Bi_{0.85}FeO_3$ films were subsequently grown at 700 °C under 0.15 mbar oxygen partial pressure with a laser fluence of 1.14 J cm$^{-2}$ keeping the repetition rate at 8 Hz (43,200 pulses for the growth of 100-nm-thick film). Post-growth, the films were cooled down with a cooling rate of 10 °C/min. The heterostructure was transferred to a high-vacuum magnetron sputtering chamber with a base pressure of ~$10^{-7}$ mbar. The $Co_{90}Fe_{10}$ and Pt layers were deposited via DC magnetron sputtering with argon pressure ranging from $2 \cdot 10^{-3}$ to $7 \cdot 10^{-3}$ mbar under a static magnetic field of 200 Oe. The electrodes were patterned by photolithography and argon-ion milling. The samples used were kept in ambient conditions.

### Scanning-probe microscopy

Scanning-probe microscopy measurements were conducted using an NTEGRA Prima scanning-probe microscope (NT-MDT Spectrum Instruments) and a Bruker Multimode 8 atomic force microscope. To apply force to the surface of the sample, we used diamond-coated DCP20 tips from NT-MDT. The force was calibrated using a force-distance curve, see supplementary information Fig. S14. Topography and PFM were performed with μmasch HQ:NSC35/Pt tips in contact mode. During raster-scanning, a 3-V peak-to-peak AC voltage modulation was applied to the tip at ~70 kHz. Ferroelectric poling was induced by applying a DC bias of 8 V to the tip. The bottom $SrRuO_3$ electrode was grounded. The same test region for the stress treatment and electric poling is identified and overlaid using the characteristic topography change accompanying the polar-to-antipolar phase transition. In addition, the clear drop of piezoresponse in the antipolar region enables a convenient location of the stressed regions. During the different poling or local pressure application, the sample remains fixed in the AFM. The PFM images in Figs. 1–4 were recorded simultaneously in Cartesian coordinates (using $X$ and $Y$ outputs of the lock-in amplifiers rather than $R$ and $\theta$). This way, instrumental background noise interfering with the measurements was minimized. The PFM switching spectroscopy loops were measured using an Asylum Research AFM system (MFP-3D) with HQ:NSC18/Pt tips (MicroMasch). Voltage pulses of 12.5 ms duration with incrementally increasing amplitude were applied to the tip. The PFM signal was measured in the resonance-enhanced PFM mode at an AC modulation frequency of 350 kHz and amplitude of 0.6 V.

### X-ray diffraction techniques

X-ray symmetric $\theta - 2\theta$ scan and reciprocal space mapping were performed using a Panalytical X'Pert3 MRD four-circle diffractometer at a wavelength of 1.5406 Å. X-ray reflectivity was employed to measure the thickness of the thin films.

### HAADF-STEM

Cross-sectional specimens of single-phase films or across different phases were prepared for transmission electron microscopy with an FEI Helios NanoLab 600i focused ion beam (FIB) instrument operated at accelerating voltages of 30 and 5 kV. Scanning transmission electron microscopy (STEM) imaging was attained with a probe-corrected FEI Titan Themis microscope operated at 300 kV. Atomic-resolution imaging of the $La_{0.15}Bi_{0.85}FeO_3$ thin films was performed by high-angular dark-field (HAADF) STEM, whose signal is proportional to $Z^n$ ($n \approx 1.6$–2.0), with Z as the atomic number. A probe convergence semi-angle of 18 mrad and collecting semi-angles of 70–190 mrad for the HAADF-STEM detector were used.

To correct for the scan distortions, time series consisting of 10 frames (2048 × 2048 pixels) were acquired and averaged by rigid and non-rigid registration using the Smart Align software[59]. The processing of the resulting HAADF-STEM images was performed in MATLAB, using custom-developed scripts as follows. First, the raw data were background-corrected and denoised using the procedure described in ref. 60. Subsequently, the atomic column positions in the corrected images were fitted by means of a center-of-mass peak-finding algorithm and refined by solving a least-squares minimization problem using the Levenberg–Marquardt algorithm. This iterative refinement makes use of seven-parameter two-dimensional Gaussians. The fitting allows quantitative estimation of the atomic column peak intensities and their positions with picometer precision[61,62]. For comparison, the fitted intensities of the $Bi^{3+}/La^{3+}$ atomic columns in each image were normalized to the maximum atomic-column intensity. Finally, a quantitative analysis of the lattice parameters was performed by means of a peak-pair analysis[62].

### Optical SHG

130-fs laser pulses with a repetition rate of 1 kHz and a wavelength of 1400 nm were used for all SHG measurements. To achieve maximum intensity, we aligned the polarization of the fundamental laser pulse along $[1\bar{1}0]_o$ using a half-wave plate and probed the frequency-doubled light along the same axis using a Glan-Taylor prism. We detected the frequency-doubled light with a liquid-nitrogen-cooled charge-coupled device (CCD) camera. A spatial resolution of ~3 μm was achieved with a long-working-distance microscope objective. All experiments were conducted in a normal-incidence geometry to minimize optical SHG contributions from the surface or interfaces.

## Data availability

The source data that supports the findings of this study are provided with this paper. Source data are provided with this paper.

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

## Acknowledgements

The authors acknowledge T. Scharsach for assistance with scanning-probe microscopy. M.T. and B.Y. acknowledge the ETH Zurich Research Grant funding under reference. 22-2 ETH-016 and the Swiss National Science Foundation under project no. 200021_188414 and 200021_231428. M.T. and M.F. acknowledge support by the EU European Research Council under the Advanced Grant Program no. 694955-INSEETO. M.D.R. acknowledges support by the Swiss National Science Foundation under project no. 200021_175926. R.R. acknowledges the support of the Army Research Laboratory and was accomplished under Cooperative Agreement Number W911NF-24-2-0100. We thank S. Reitz and J. Hecht for technical support. We thank S. Vélez for support with the lithography. We further thank B. Grosso and N.A. Spaldin for fruitful discussions.

## Author contributions

M.M. initiated and designed the project with M.T. and M.D.R. Furthermore, M.M., H.K. and B.Y. performed and analyzed the scanning-probe measurements. H.L. and A.G. performed the PFM switching spectroscopy characterization. M.M. performed and analyzed the optical SHG measurements. M.D.R. performed and analyzed the HAADF-STEM measurements. B.Y. and Y.-L.H. grew the samples under the supervision of M.T. and R.R., respectively. M.M. deposited the top electrodes of the capacitors. M.T. supervised the work jointly with M.F. The manuscript was written by M.M., B.Y., M.D.R., M.F. and M.T. with input from all authors.

## Competing interests

The authors declare no competing interests.
