## [Transparent Peer Review file · Nature Communications]

Reversible control over the distribution of chemical inhomogeneities in multiferroic BiFeO₃

Corresponding Author: Professor Morgan Trassin

Version 0:

Reviewer comments:

Reviewer #1

(Remarks to the Author)

This manuscript presents a force- and composition-driven phase transition in multiferroic La doped BiFeO₃ thin films. The polar and anti-polar phase can be reversibly controlled. This study is interesting. The manuscript is convincing and well-written. I recommend it for publication in Nature Communications after considering my following concerns.

1. The authors shows that in the as-grown La-BiFeO₃ , the distribution of La³⁺ is homogeneous. However, the strain drives the inhomogeneous distribution, giving rise to the polar to anti-polar phase transition. The authors should give more explanations of this phenomena.
2. XRD data would give more information of the phase structure.
3. The mismatch strain between the film and substrate is a way to induce strain as well. Why does the authors choose DSO as the substrate? What will happen if STO or other substrates were used?
4. In addition, why was 15% La doping BiFeO₃ chosen to be studied? Should it be the same results using 10% , 20% La doped BiFeO₃ target?

Reviewer #2

(Remarks to the Author)

Authors have reported the influence of the local arrangement of chemical inhomogeneities on the material's functionalities is underexplored. For this purpose, they have chosen a very well-known multiferroic material. I have several questions and comments about the work which are given below:

Q1: It is state: "We thus conclude that the applied compressive force triggered an alternating enrichment and depletion of the La³⁺ distribution in our films by using diamond coated scanning probe tip". What are the dimensions of this tip? And how was the same area for measurements chosen for AFM studies?

Q2: What are expected or reported results if we replace the DyScO₃ with SrTiO₃? This is important as the lattice mismatch is quite different when compared with the lattice parameters of BFO. This comparison is important before using rare substrate like DyScO₃.

Q3: In line 96, the author reports that VPFM indicates single polarization domain while LPFM indicates four in-plane Polarization domains. How does the author identify the presence of four different domains from the figure of LPFM? And how do they argue on the bases color contrast that they have single domain in VPFM?.

Q4: What is the geometry of the system adopted to take the LPFM images? Means how was sample polarized by the application of electric field in the lateral direction?

Q5: In Figure 2(a) author have shown the presence of two different La/Bi layers in which one layer atom show an up down shift while the other layer's atom remains on their original position. The atomic lines which undergo a shift are identified as

La₃₊ rich layer while the second one which remains intact are labeled as La₃₊ poor layer. Why atomic displacements are observed in one layer and not in the other? And why the line which shows displacement are labeled as La₃₊ rich layers? I would like to see the explanation of such observation; is it has to do with e.g. the bond strength difference of La compared to Bi?

Q6: It is reported that "Therefore, we applied a static electric field with a biased scanning-probe tip to a small square within the force generated anti-polar region". This selected region is of very small dimensions of the sample then how does author identified this region again and apply electric field on the similar region?

Q7: With an applied voltage of 8 V the coercive field of the ferroelectric phase is exceeded by a factor of two. How does the author calculate corecivity? How is it identified that coercive field increase by the factor of two? And why does it increases by the factor of two? Did you separately measure complete hysteresis for this?

Q8: It is stated that "We deposited circular Co₉₀Fe₁₀/Pt top electrodes with a layer thickness of 2 nm, hence creating a SrRuO₃/La_{0.15}Bi_{0.85}FeO₃/Co₉₀Fe₁₀ capacitor". This unclear that whether you had Pt or Co₉₀Fe₁₀ as top electrode? In addition to that what is the purpose of using CoFe alloy instead of Pt as electrode?

Q9: It is reported that "Our results, depicted in supplementary information, Fig. S5, show that the corrugation always aligns with respect to the [110]o-axis of the DyScO₃ substrate, independent of the diamond-tip scanning direction." As SrRuO₃ layer is grown on the top of DyScO₃ then how DyScO₃ affects the growth of crystal orientation of substituted BFO layer? In addition to that the thickness of the film is 100nm. Isn't that 100nm thickness is large and for such study where role of DyScO₃ is claimed to be important, ultrathin films of BiLaFeO₃ should prepared. For example, if the film thickness is around 10nm then it will be much better to study such a system and see the effect of strain induced by the substrate.

Q10: Why pure BFO system is not selected for this analysis? As BFO undergoes phase transformation by applying external pressure then why pure BFO system is not selected for this analysis? Strain induced by the substrate can be utilized for the phase transition in BFO.

Q11: Title of the paper is required to be revised which indicates that the results are of general importance however the study is about a specific material prepared in specific conditions and a particular geometry is used.

Q12: BFO is a well known and well-studied material. I would like to see a clear and concise novelty state of the present work for publication. For publication what is the clear significance of the present work.

Reviewer #3

(Remarks to the Author)

The paper discusses the control over the distribution of chemical inhomogeneities with the example of the oxide thin film La-BiFeO₃ (BFO). With applying force and electrical field, the pristine thin films were modified with the result to change between ferroelectric and anti-polar phases. This concept was not only shown in model samples, but was also transferred to a device-like sample to demonstrate its application feasibility.

It is a fascinating paper, inspiring topic and results! Congratulations! Next to the nice research, I also appreciate the well-worked out and written manuscript and the clear structure of the manuscript. It was a pleasure to read the manuscript! Still, I have a few questions, remarks, discussion points that need to be considered before publication of the manuscript. In particular, a few conclusions are too fast in my view (see detailed comments below).

I want to mention that I am not an expert in ferroelectrics and I suggest that a person with this expertise should also review this manuscript. I suggest a few reviewers to the editor directly. From my side, I support that this manuscript is considered for publication in Nature Communications after considering the comments below.

Comments:

- I am missing XRD and XPS data that show the phases of the thin film and the composition of the top layer of the thin film, respectively. In particular, I am interested whether the top surface has the same composition as the bulk of the thin film prior to any measurement and preferably also after the measurements; often this is not the case. In particular in BFO materials, the top surface can also be terminated with a Bi-oxide rich layer.

- P. 2, l. 70: Can you please motivate why you choose this exact stoichiometry: La_{0.15}Bi_{0.8}FeO₃; why do you choose for this ratio of La and Bi?

- I am missing some proof of your layer thicknesses (p. 2, line 73). Do you report targeted thin film thicknesses or real thicknesses, e.g. measured by SEM cross section or TEM?

- Did you check the final stoichiometry of La_{0.15}Bi_{0.8}FeO₃? Or is this the stoichiometry of the PLD target?

- P.3, text and Fig. 1: the following is not clear

o What does Bi / La in Fig. 1 mean? Either Bi or La OR the ratio of Bi and La?

o Normalized intensity – you refer to the intensity of what? Be more specific here

o L. 84: "random distribution of La" – how can you conclude from the image to a random distribution of La, you also have Bi. Where does the chemical specific information about Bi and La come in?

o L. 84 you talk about "random distribution of La" and on l. 92 you talk about "homogeneous La distribution" – this is confusing

- P. 3, l. 95: "atomically flat" – an atom is about 1 Angstrom in size. The AFM image has a micron bar of 2 micrometer. In the AFM image, quite some roughness is seen. But more important is that in the resolution of the AFM image with 2 micron bar, you cannot see atomical flatness – higher resolution AFM is needed for this conclusion.

- P.3, l. 97: "four distinct piezo response levels" – how do I see this?

- P. 5, first paragraph:

o l. 128: Also here it is not clear to me, how you can conclude from your data about the chemistry; how do you know that it is La and not Bi?

o From Fig. 2 and the discussion around, I understand that you observe to things: 1) movement of atoms up and down (red, blue arrows), 2) different intensity compared to Fig. 1; the movement (1) I believe is due to the pressure from the top, the atoms move sideways – correct?; the different intensity is not fully clear to me – hence, why do you get different intensity?

Because atoms move in different layers?, what is the model here – in general, these 2 movements of the atoms need to be better separately discussed with a model.

- P. 6: very nice that the process is reversible. But what about (self-)relaxation? How long do the changes that you imply by force or electrical field stay, if you do not do anything?

- You very often write “we” and “our” – this is not forbidden, but I would not use too much in scientific writing.

- Methods – Sample fabrication, p. 14: this paragraph is poor and additional information is needed; probably the other methods sections need to be revised similarly, but this needs to be done by specialists of the respective techniques – I do not comment on these. Questions for sample fabrication are among others:

o Which instrument did you use?

o In which lab where the samples prepared?

o Which type of laser was used for the PLD

o How many pulses? – it should be possible to calculate a deposition rate with the information in the paper; and/or the deposition rate should be mentioned directly

o What thickness is targeted?

o Was the thickness calibrate prior to the measurements?

o Repetition rate of laser pulses?

o What was the cooling rate?

o Where the samples freshly deposited samples? How were samples stored before measurement?

o Information about the electrode fabrication is missing fully.

o It is not mentioned in the author contribution or acknowledgement who did the thin film depositions and the electrode fabrication.

- Supplementary text:

o Fig. S1: I think the authors do not refer to this figure in the main text.

o Fig. S3: details about the simulations would be helpful.

o Fig. S6: in the text of the subtitle words seem to be missing.

Anja Bieberle

Version 1:

Reviewer comments:

Reviewer #1

(Remarks to the Author)

The revised manuscript has been improved according to my suggestions.

Reviewer #2

(Remarks to the Author)

Authors have addressed questions raised during first cycle. This work can be considered for publication.

Reviewer #3

(Remarks to the Author)

The authors carefully worked on the comments in my first review. They answered to all my questions both in the document itself as well as in the manuscript. The authors added a lot of new information and measurements and explained this will. I am very happy how the authors did this and the manuscript has improved considerably. I recommend the manuscript to be published now.

Response to reviewers

"Reversible control over the distribution of chemical inhomogeneities in multiferroic BiFeO₃"

M. Müller, B. Yan, H. Ko, Y.-L. Huang, H. Lu, A. Gruverman, R. Ramesh, M. D. Rossell, M. Fiebig, and M. Trassin
(Manuscript ID NCOMMS-24-43112)

We thank all the reviewers for their careful reading of the manuscript. We appreciate their very encouraging feedback and the enthusiastic recommendation of the publication of our work once revisions are performed. We have now addressed the reviewers' comments, performed all the suggested measurements, and revised the manuscript and the Supplementary Information accordingly. In particular, we now provide insights into the ionically active functionality in our films and provide more information on the methods employed. The revisions have involved new collaborators and group members, as reflected in the revised author list. Our point-by-point response to the reviewers' comments (shown in blue, italics) is provided below. Numbered references correspond to the bibliography of the main text.

Reviewer #1:

This manuscript presents a force- and composition-driven phase transition in multiferroic La doped BiFeO₃ thin films. The polar and anti-polar phase can be reversibly controlled. This study is interesting. The manuscript is convincing and well-written. I recommend it for publication in Nature Communications after considering my following concerns.

We thank the reviewer for the encouraging feedback. We address the reviewer's concerns in the following.

1.1. The authors show that in the as-grown La-BiFeO₃, the distribution of La³⁺ is homogeneous. However, the strain drives the inhomogeneous distribution, giving rise to the polar to anti-polar phase transition. The authors should give more explanations of this phenomena.

Our work is motivated by the recent reports highlighting correlations between lattice chemistry and polarization in epitaxial systems (43,44,45) in which tuning the concentration and distribution of ions provides an additional degree of freedom to control application-relevant functionalities.(16)

In ferroelectric BiFeO₃, the Bi³⁺ electronic lone pair drives local polar distortions in the ferroelectric unit cell in BiFeO₃.(24,25,26) Hence, the Bi³⁺ substitution with La³⁺ influences the net polar state in the La-BiFeO₃ films. At the pressure-induced polar-to-antipolar phase transition in highly La³⁺ substituted BiFeO₃ thin films, the reorganization of the atomic displacement within the unit cell may, hence, correlate with a change in lattice chemistry. Here, the identical radii and valence states of Bi³⁺ and La³⁺ cations at play most likely facilitate the reversible cationic redistribution across the phase transition, as highlighted in the report of the different polar regions in the phase diagram of BiFeO₃/LaFeO₃ superlattices.(49) We have extended this discussion in the main text, page 6.

1.2. XRD data would give more information of the phase structure.

As requested, we have added the XRD data in the supplementary information, Fig. S1. We also provide an extended electron microscopy-based analysis of the pressure-induced antipolar phase structure, Fig. S6.

1.3. The mismatch strain between the film and substrate is a way to induce strain as well. Why does the authors choose DSO as the substrate? What will happen if STO or other substrates were used?

We selected DSO substrates motivated by its use for the seminal demonstration of multiferroic and magnetoelectric engineering in the BiFeO₃ system.(20,38) As suggested by the referee, we grew additional films and now provide evidence of the pressure-induced polar to antipolar phase transition in films grown on STO. The corresponding AFM and PFM scans are shown in Fig. S12. The impact of the substrate is now discussed in the main text, page 11.

1.4. In addition, why was 15% La doping BiFeO3 chosen to be studied? Should it be the same results using 10%, 20% La doped BiFeO3 target?

Our choice of La³⁺ substitution level is motivated by bulk studies on ceramics revealing a morphotropic phase boundary between polar and non-polar phases at 15% La³⁺ content.(37) We now clarify this in the revised main text, page 3.

Motivated by the reviewer's suggestion, we investigated the impact of the level of La³⁺ substitution. The pressure-induced phase transition is only achieved for 10% and 15 % substitution levels. Films with 20% La³⁺ substitution exhibit the non-polar phase in the pristine state. The ferroelectric phase cannot be triggered by an external electric field. Films without La³⁺ substitution exhibit a ferroelectric order in the pristine state, and the application of local pressure does not trigger the emergence of the antipolar phase.

We added the corresponding AFM and PFM scans in supplementary information, Fig. S9 and S10. The influence of the La³⁺ substitution level is now discussed in the main text, page 8.

Reviewer #2:

Authors have reported the influence of the local arrangement of chemical inhomogeneities on the material's functionalities is underexplored. For this purpose, they have chosen a very well-known multiferroic material. I have several questions and comments about the work which are given below:

2.1. It is state: "We thus conclude that the applied compressive force triggered an alternating enrichment and depletion of the La³⁺ distribution in our films by using diamond coated scanning probe tip". What are the dimensions of this tip? And how was the same area for measurements chosen for AFM studies?

For our study, we have been using diamond tips (models DCP20 and DT-NCHR-20 from μ masch) with a tip radius around 100 nm.

The spatial correlation between the local pressure and piezoresponse and AFM measurements is enabled by scanning tip exchange while keeping the sample fixed. After pressure application using a diamond tip, we use Pt coated silicon tips (HQ:NSC35/Pt, μ masch) for AFM and PFM. We take advantage of the clear topography change upon the phase transition that allows convenient location of the pressure-induced phase transition. We now provide all this information in the methods section.

2.2. What are expected or reported results if we replace the DyScO3 with SrTiO3? This is important as the lattice mismatch is quite different when compared with the lattice parameters of BFO. This comparison is important before using rare substrate like DyScO3.

We agree. We performed the suggested experiments, please refer to 1.3. We now report the pressure-induced phase transition in films grown on SrTiO₃ in the revised manuscript.

2.3. In line 96, the author reports that VPFM indicates single polarization domain while LPFM indicates four in-plane Polarization domains. How does the author identify the presence of four different domains from the figure of LPFM? And how do they argue on the bases color contrast that they have single domain in VPFM?

We understand the confusion caused by an issue with the scaling of the piezoresponse in the original figure. We now provide revised LPFM and VPFM data in Fig. 1, taken on a sample exhibiting larger domains in the pristine state. In addition, we performed local out-of-plane electric field poling using the scanning probe tip and determine the initial out-of-plane polarization component based on the VPFM.

2.4. What is the geometry of the system adopted to take the LPFM images? Means how was sample polarized by the application of electric field in the lateral direction?

We now indicate the crystal orientation of sample during the PFM scan in Fig. 1 and 2. The cantilever and slow-axis directions are detailed in the figure captions. We did not observe an impact of the in-plane direction on electrical poling or pressure-induced antipolar phase conversion, as discussed in supplementary information Fig. S5.

2.5. In Figure 2(a) author have shown the presence of two different La/Bi layers in which one layer atom show an up down shift while the other layer's atom remains on their original position. The atomic lines which undergo a shift are identified as La³⁺ rich layer while the second one which remains intact are labeled as La³⁺ poor layer. Why atomic displacements are observed in one layer and not in the other? And why the line which shows displacement are labeled as La³⁺ rich layers? I would like to see the explanation of such observation; is it has to do with e.g. the bond strength difference of La compared to Bi?

The atomic displacements within every other plane in the antipolar phase are a characteristic feature of the antipolar *Pnma* phase.⁽⁴⁶⁾ We now highlight this better in the revised manuscript, page 6. Electron microscopy analysis could further correlate the up/down shift in the atomic plane with a local variation of the La³⁺ content. In addition to our STEM-based contrast (sensitive to atomic number) we provide additional EDX and EELS data demonstrating that the up/down shifts take place in atomic planes with a reduced amount of La³⁺, see Fig. S2. Note that solely the Bi³⁺ electronic lone pair drives local polar distortions in the unit cell.⁽²⁵⁾ We discuss the possible explanation of such an observation regarding the correlations between lattice chemistry and polarization in epitaxial systems, please refer to our answer 1.1. We have also extended this discussion in the main text, page 6.

2.6. It is reported that "Therefore, we applied a static electric field with a biased scanning-probe tip to a small square within the force generated anti-polar region". This selected region is of very small dimensions of the sample then how does author identified this region again and apply electric field on the similar region?

We can localize the antipolar phase induced after local pressure application thanks to the associated topography change. Prior to the electric-field application, the pressure-induced antipolar phase region is identified using AFM. Localization is further aided by not moving the sample between exerting the pressure and the electric field. We now provide this information in the methods section.

2.7. With an applied voltage of 8 V the coercive field of the ferroelectric phase is exceeded by a factor of two. How does the author calculate corecivity? How is it identified that coercive field increase by the

factor of two? And why does it increases by the factor of two? Did you separately measure complete hysteresis for this?

In the original manuscript, we applied a poling-tip bias of 8 V (twice the voltage needed for local ferroelectric switching in pristine regions) to ensure the field-induced antipolar-to-polar transition. Thanks to collaboration with PFM experts, we now provide a quantitative analysis of the local electric-field-induced phase transition. The application of a mere 2 V tip bias suffices to recover the ferroelectric state from the antipolar state. The local piezoresponse spectroscopy measurements in the polar and antipolar pristine regions are shown in the main text, Fig. 5. We edited the text accordingly, page 9.

2.8. It is stated that “We deposited circular Co90Fe10/Pt top electrodes with a layer thickness of 2 nm, hence creating a SrRuO3/La0.15Bi0.85FeO3/Co90Fe10 capacitor”. This unclear that whether you had Pt or Co90Fe10 as top electrode? In addition to that what is the purpose of using CoFe alloy instead of Pt as electrode?

We do have Pt(2 nm)/CoFe(2 nm) top electrodes. We corrected the main text, thank you. The use of such capping layers in seminal magnetoelectric multiferroic BiFeO₃-based device implementation (20,27,38,55,56) motivates our selection. Hence, we underline the application relevance of the proof-of-concept architecture. We added this information in the main text, page 9.

2.9. It is reported that “Our results, depicted in supplementary information, Fig. S5, show that the corrugation always aligns with respect to the [110]o-axis of the DyScO3 substrate, independent of the diamond-tip scanning direction.” As SrRuO3 layer is grown on the top of DyScO3 then how DyScO3 affects the growth of crystal orientation of substituted BFO layer? In addition to that the thickness of the film is 100nm. Isn't that 100nm thickness is large and for such study where role of DyScO3 is claimed to be important, ultrathin films of BiLaFeO3 should prepared. For example, if the film thickness is around 10nm then it will be much better to study such a system and see the effect of strain induced by the substrate.

The 100-nm-thick films are coherently strained to the DyScO₃ substrates, hence the correlation between the substrate crystal structure and the La_{0.15}Bi_{0.85}FeO₃ orientation is maintained. We now provide in supplementary information Fig. S1 the XRD analysis and the reciprocal space mapping of the La_{0.15}Bi_{0.85}FeO₃ film and refer to XRD in the revised main text, page 3.

Motivated by the reviewer's suggestion, we also studied the behavior of sub-100-nm layers. The domain size shrinks down below our scanning-probe tip resolution in the ultrathin regime, we provide however the evidence of reversible pressure-induced phase polar-to-antipolar phase transition in 20-nm-thick layers in supplementary Fig. S13. We discuss this last point in the main text, page 11.

2.10. Why pure BFO system is not selected for this analysis? As BFO undergoes phase transformation by applying external pressure then why pure BFO system is not selected for this analysis? Strain induced by the substrate can be utilized for the phase transition in BFO.

We have addressed the impact of the La³⁺ substitution in the revised manuscript. Please refer to 1.4. We do not observe the pressure-induced phase transition in pure BiFeO₃ films. A La³⁺ amount of at least 10% is needed to bring the system close to the phase transition. This point is now discussed in the supplementary information, see Fig. S9 and in the main text, page 8.

2.11. Title of the paper is required to be revised which indicates that the results are of general importance however the study is about a specific material prepared in specific conditions and a particular geometry is used.

We edited the title according to this recommendation.

2.12. BFO is a well-known and well-studied material. I would like to see a clear and concise novelty state of the present work for publication. For publication what is the clear significance of the present work.

The rich phase diagram of multiferroic BFO has motivated our material selection for this study. As suggested, we added a concise statement on the novelty of our work in the conclusion section, opening new opportunities for multiferroic phase engineering using pressure-induced chemical inhomogeneity distribution.

Reviewer #3:

The paper discusses the control over the distribution of chemical inhomogeneities with the example of the oxide thin film La-BiFeO₃ (BFO). With applying force and electrical field, the pristine thin films were modified with the result to change between ferroelectric and anti-polar phases. This concept was not only shown in model samples, but was also transferred to a device-like sample to demonstrate its application feasibility.

It is a fascinating paper, inspiring topic and results! Congratulations! Next to the nice research, I also appreciate the well-worked out and written manuscript and the clear structure of the manuscript. It was a pleasure to read the manuscript!

Still, I have a few questions, remarks, discussion points that need to be considered before publication of the manuscript. In particular, a few conclusions are too fast in my view (see detailed comments below). I want to mention that I am not an expert in ferroelectrics and I suggest that a person with this expertise should also review this manuscript. I suggest a few reviewers to the editor directly. From my side, I support that this manuscript is considered for publication in Nature Communications after considering the comments below.

We thank the reviewer for the encouraging feedback. We address the reviewer's concerns in the following.

3.1. I am missing XRD and XPS data that show the phases of the thin film and the composition of the top layer of the thin film, respectively. In particular, I am interested whether the top surface has the same composition as the bulk of the thin film prior to any measurement and preferably also after the measurements; often this is not the case. In particular in BFO materials, the top surface can also be terminated with a Bi-oxide rich layer.

We fully agree and took STEM cross-sectional images of the top surface of the films in the pristine ferroelectric region and after the local pressure-induced antipolar phase transition. We added this data in Supplementary Fig. S6. We do not observe any surface reconstruction after the measurements. This is now mentioned in the main text, page 6.

3.2. P. 2, l. 70: Can you please motivate why you choose this exact stoichiometry: La_{0.15}Bi_{0.8}FeO₃; why do you choose for this ratio of La and Bi?

Our choice of La³⁺ substitution level is motivated by bulk studies on ceramics revealing a morphotropic phase boundary at 15% La³⁺ content.⁽³⁷⁾ We now clarify this in the revised main text, page 3.

3.3. I am missing some proof of your layer thicknesses (p. 2, line 73). Do you report targeted thin film thicknesses or real thicknesses, e.g. measured by SEM cross section or TEM?

Film thickness was determined by combining XRD and X-ray reflectivity and low-magnification electron-microscopy cross-section imaging. We provide a representative thickness-determination measurement in supplementary information S1, and added the information in the methods section and refer to it in the main text, page 3.

3.4. Did you check the final stoichiometry of La_{0.15}Bi_{0.8}FeO₃? Or is this the stoichiometry of the PLD target?

We verified the composition of the films by STEM. The 15% La³⁺ substitution in the target is maintained in the film. We provide the results of the chemical analysis in the supplementary information, table S1.

3.5. P.3, text and Fig. 1: the following is not clear. What does Bi / La in Fig. 1 mean? Either Bi or La OR the ratio of Bi and La? Normalized intensity – you refer to the intensity of what? Be more specific here

We provide the missing information in the figure captions of the revised manuscript. With Bi³⁺/La³⁺ we refer to the planes where Bi³⁺ and La³⁺ occupy the A-sites. We refer to the STEM intensity normalized to the maximum intensity. More information is provided in the methods section.

3.6. L. 84: “random distribution of La” – how can you conclude from the image to a random distribution of La, you also have Bi. Where does the chemical specific information about Bi and La come in? L. 84 you talk about “random distribution of La” and on l. 92 you talk about “homogeneous La distribution” – this is confusing

We apologize for the confusion; we meant homogenous distribution. We corrected the main text.

3.7. P. 3, l. 95: “atomically flat” – an atom is about 1 Angstrom in size. The AFM image has a micron bar of 2 micrometer. In the AFM image, quite some roughness is seen. But more important is that in the resolution of the AFM image with 2 micron bar, you cannot see atomical flatness – higher resolution AFM is needed for this conclusion.

All our films exhibit unit-cell height topography. We now refer to unit-cell height topography in the main text.

3.8. P.3, l. 97: “four distinct piezo response levels” – how do I see this?

We replaced the PFM images for a more convenient visualization of the contrast levels in figure 1. We do observe contrast levels in the LFPM data corresponding to the characteristic in-plane domain states associated with four different polarization directions.(41) We corrected the main text, page 4.

3.9. P. 5, first paragraph: l. 128: Also here it is not clear to me, how you can conclude from your data about the chemistry; how do you know that it is La and not Bi?

The STEM contrast depends on the atomic number, and hence, we can conclude on the relative Bi³⁺/La³⁺ ratio in each atomic column. We added this information in the main text of the revised manuscript, page 3. Furthermore, we provide additional chemical analysis in supplementary information, Fig. S2. This further confirms the variation in La³⁺ content in the different atomic planes.

3.10. From Fig. 2 and the discussion around, I understand that you observe to things: 1) movement of atoms up and down (red, blue arrows), 2) different intensity compared to Fig. 1; the movement (1) I

believe is due to the pressure from the top, the atoms move sideways – correct?; the different intensity is not fully clear to me – hence, why do you get different intensity? Because atoms move in different layers?, what is the model here – in general, these 2 movements of the atoms need to be better separately discussed with a model.

Let us first clarify the movement of atoms up and down (1). This up-and-down atomic displacement of the A cation in every other plane in the antipolar phase is a characteristic feature of the antipolar *Pnma* phase (ref 46). We now highlight this better in the revised manuscript, page 6. The phase transition is triggered by local pressure application. We find that while in the polar phase the La^{3+} is homogeneously distributed in the lattice, there is a clear ordering of the La^{3+} occupation sites within the atomic planes in the antipolar phase. The difference in atomic number between La and Bi results in a clear STEM intensity contrast (2) within the planes labeled as La^{3+} -poor and La^{3+} -rich (2).

Our experimental results hence suggest a striking correlation between the lattice chemistry and the polar nature of the unit cell. Please refer to 1.1. for more detailed discussion. We have extended this discussion in the main text, page 6. We hope this experimental study emerges as a wake-up call triggering A-site cation occupancy reconfiguration investigation in functional perovskites, as first pointed out in ref 48.

3.11. P. 6: very nice that the process is reversible. But what about (self-)relaxation? How long do the changes that you imply by force or electrical field stay, if you do not do anything?

The patterns we have created by pressure or electric field are stable for at least several months. In fact, we have not seen any signs of degradation since the first evidence of the local phase transition. We now provide in the supplementary AFM and PFM scans of a region images three months after local electric field and pressure application, see Fig. S11. The stability of the phase transition over time is now mentioned in the main text, page 9.

3.12. You very often write “we” and “our” – this is not forbidden, but I would not use too much in scientific writing.

We now refrain from writing “we” and “our” in the main text.

3.13. Methods – Sample fabrication p. 14: this paragraph is poor and additional information is needed; probably the other methods sections need to be revised similarly, but this needs to be done by specialists of the respective techniques – I do not comment on these. Questions for sample fabrication are among others: Which instrument did you use? In which lab where the samples prepared? Which type of laser was used for the PLD How many pulses? – it should be possible to calculate a deposition rate with the information in the paper; and/or the deposition rate should be mentioned directly. What thickness is targeted? Was the thickness calibrate prior to the measurements? Repetition rate of laser pulses? What was the cooling rate? Where the samples freshly deposited samples? How were samples stored before measurement? Information about the electrode fabrication is missing fully.

The $\text{La}_{0.15}\text{Bi}_{0.85}\text{FeO}_3/\text{SrRuO}_3$ films were grown on single-crystalline (110)_o-oriented DyScO_3 substrates (CrysTec GmbH) by pulsed laser deposition using a 248 nm KrF excimer laser at ETH Zurich and University of California, Berkeley. The SrRuO_3 buffer layer was deposited at 700°C under 0.15 mbar oxygen partial pressure with a laser fluence of 0.69 J cm⁻² and a laser repetition rate of 8 Hz (9600 pulses for the growth of 14 nm-thick film). The $\text{La}_{0.15}\text{Bi}_{0.85}\text{FeO}_3$ films were subsequently grown at 700°C under 0.15 mbar oxygen partial pressure with a laser fluence of 1.14 J cm⁻² keeping the repetition rate at 8 Hz (43200 pulses for the growth of 100 nm-thick film). Post-growth, the films were cooled down with a cooling rate of 10°C/min. The samples used were kept in ambient conditions. The $\text{Co}_{90}\text{Fe}_{10}$ and Pt layers were deposited via DC magnetron sputtering with argon pressure ranging from 2·10⁻³ to

$7 \cdot 10^{-3}$ mbar under a static magnetic field of 200 Oe. The electrodes were patterned by photolithography and argon-ion milling.

We added all this information in the methods section of the revised manuscript.

3.14. It is not mentioned in the author contribution or acknowledgement who did the thin film depositions and the electrode fabrication.

We edited this section to better reflect the authors' contributions.

3.15. Supplementary text: Fig. S1: I think the authors do not refer to this figure in the main text. Fig. S3: details about the simulations would be helpful. Fig. S6: in the text of the subtitle words seem to be missing.

We refer to it in the method section. The sequence of figures is corrected in the supplementary information.

The paper discusses the control over the distribution of chemical inhomogeneities with the example of the oxide thin film La-BiFeO₃ (BFO). With applying force and electrical field, the pristine thin films were modified with the result to change between ferroelectric and anti-polar phases. This concept was not only shown in model samples, but was also transferred to a device-like sample to demonstrate its application feasibility.

It is a fascinating paper, inspiring topic and results! Congratulations! Next to the nice research, I also appreciate the well-worked out and written manuscript and the clear structure of the manuscript. It was a pleasure to read the manuscript!

Still, I have a few questions, remarks, discussion points that need to be considered before publication of the manuscript. In particular, a few conclusions are too fast in my view (see detailed comments below).

I want to mention that I am not an expert in ferroelectrics and I suggest that a person with this expertise should also review this manuscript. I suggest a few reviewers to the editor directly. From my side, I support that this manuscript is considered for publication in Nature Communications after considering the comments below.

Comments:

- I am missing XRD and XPS data that show the phases of the thin film and the composition of the top layer of the thin film, respectively. In particular, I am interested whether the top surface has the same composition as the bulk of the thin film prior to any measurement and preferably also after the measurements; often this is not the case. In particular in BFO materials, the top surface can also be terminated with a Bi-oxide rich layer.
- P. 2, l. 70: Can you please motivate why you choose this exact stoichiometry: La_{0.15}Bi_{0.8}FeO₃; why do you choose for this ratio of La and Bi?
- I am missing some proof of your layer thicknesses (p. 2, line 73). Do you report targeted thin film thicknesses or real thicknesses, e.g. measured by SEM cross section or TEM?
- Did you check the final stoichiometry of La_{0.15}Bi_{0.8}FeO₃? Or is this the stoichiometry of the PLD target?
- P.3, text and Fig. 1: the following is not clear
 - o What does Bi / La in Fig. 1 mean? Either Bi or La OR the ratio of Bi and La?
 - o Normalized intensity – you refer to the intensity of what? Be more specific here
 - o L. 84: “random distribution of La” – how can you conclude from the image to a random distribution of La, you also have Bi. Where does the chemical specific information about Bi and La come in?
 - o L. 84 you talk about “random distribution of La” and on l. 92 you talk about “homogeneous La distribution” – this is confusing
- P. 3, l. 95: “atomically flat” – an atom is about 1 Angstrom in size. The AFM image has a micron bar of 2 micrometer. In the AFM image, quite some roughness is seen. But more important is that in the resolution of the AFM image with 2 micron bar, you cannot see atomical flatness – higher resolution AFM is needed for this conclusion.
- P.3, l. 97: “four distinct piezo response levels” – how do I see this?
- P. 5, first paragraph:
 - o l. 128: Also here it is not clear to me, how you can conclude from your data about the chemistry; how do you know that it is La and not Bi?
 - o From Fig. 2 and the discussion around, I understand that you observe to things: 1) movement of atoms up and down (red, blue arrows), 2) different intensity compared

to Fig. 1; the movement (1) I believe is due to the pressure from the top, the atoms move sideways – correct?; the different intensity is not fully clear to me – hence, why do you get different intensity? Because atoms move in different layers?, what is the model here – in general, these 2 movements of the atoms need to be better separately discussed with a model.

- P. 6: very nice that the process is reversible. But what about (self-)relaxation? How long do the changes that you imply by force or electrical field stay, if you do not do anything?
- You very often write “we” and “our” – this is not forbidden, but I would not use too much in scientific writing.
- Methods – Sample fabrication, p. 14: this paragraph is poor and additional information is needed; probably the other methods sections need to be revised similarly, but this needs to be done by specialists of the respective techniques – I do not comment on these. Questions for sample fabrication are among others:
 - Which instrument did you use?
 - In which lab where the samples prepared?
 - Which type of laser was used for the PLD
 - How many pulses? – it should be possible to calculate a deposition rate with the information in the paper; and/or the deposition rate should be mentioned directly
 - What thickness is targeted?
 - Was the thickness calibrate prior to the measurements?
 - Repetition rate of laser pulses?
 - What was the cooling rate?
 - Where the samples freshly deposited samples? How were samples stored before measurement?
 - Information about the electrode fabrication is missing fully.
 - It is not mentioned in the author contribution or acknowledgement who did the thin film depositions and the electrode fabrication.
- Supplementary text:
 - Fig. S1: I think the authors do not refer to this figure in the main text.
 - Fig. S3: details about the simulations would be helpful.
 - Fig. S6: in the text of the subtitle words seem to be missing.